# Hybrid Poly(Lactic)-Chitosan Scaffold Intensifying In Situ Bioprocessing of *Rindera graeca* Transgenic Roots for Enhanced Rinderol Production

**DOI:** 10.3390/ijms262110668

**Published:** 2025-11-01

**Authors:** Kamil Wierzchowski, Szymon Bober, Aleksandra Bandzerewicz, Miroslav Šlouf, Jiří Hodan, Agnieszka Gadomska-Gajadhur, Katarzyna Sykłowska-Baranek, Maciej Pilarek

**Affiliations:** 1Faculty of Chemical and Process Engineering, Warsaw University of Technology, Waryńskiego 1, 00-645 Warsaw, Poland; szymon.bober.dokt@pw.edu.pl (S.B.); maciej.pilarek@pw.edu.pl (M.P.); 2Faculty of Chemistry, Warsaw University of Technology, Noakowskiego 3, 00-664 Warsaw, Poland; aleksandra.bandzerewicz.dokt@pw.edu.pl (A.B.); agnieszka.gajadhur@pw.edu.pl (A.G.-G.); 3Institute of Macromolecular Chemistry, Czech Academy of Sciences, Heyrovského nám. 1888/2, 162 00 Prague, Czech Republic; slouf@imc.cas.cz (M.Š.); hodan@imc.cas.cz (J.H.); 4Faculty of Pharmacy, Medical University of Warsaw, Banacha 1, 02-097 Warsaw, Poland; katarzyna.syklowska-baranek@wum.edu.pl

**Keywords:** chitosan-based biomaterial, hybrid biomaterial, in situ bioprocessing, naphthoquinones, transgenic (hairy) roots

## Abstract

In vitro cultured biomass of *Rindera graeca*, a rare endemic plant, is an efficient renewable source of bioactive naphthoquinones, e.g., rinderol, a potential bioactive inducer of apoptosis in cancer cells. Bioengineering strategies, as biomass immobilization on functionalized biomaterial-based scaffolds, elicitation by chitosan, and in situ extraction of metabolites, are tested for intensifying naphthoquinones production in *R. graeca* hairy roots. The aim of the study was to investigate the effects of hybrid poly(lactic)–chitosan scaffolds on biomass proliferation and rinderol production in *R. graeca* hairy roots. Effects of chitosan origin (fungal or squid), molecular mass (350–1800 kDa), and concentration (up to 45%) in the developed hybrid scaffolds have been quantitatively identified, and the results were compared to the reference culture system containing an unmodified PLA-based construct. Applying PLA–chitosan scaffold containing 33% of fungal chitosan resulted in 635 times higher rinderol production (3660 µg g_DW_^−1^) than the application of reference scaffolds. Among the tested parameters, the chitosan concentration in the hybrid scaffolds revealed significant importance in rinderol production. To sum up, the developed hybrid PLA-chitosan scaffold may be recognized as a functional key element supporting the production of naphthoquinones in cultures of *R. graeca* biomass.

## 1. Introduction

Plant secondary metabolites are valuable bioactive compounds whose biochemical activity and composition vary between plant species [1]. Unlike the primary ones, secondary metabolites are unnecessary for plant growth and development. However, as native biochemical agents, they are key factors in physiological functions responsible for environmental interactions and coping with abiotic and biotic stress [2]. Plant secondary metabolites find their application, mainly, as drugs in medicine, and as food additives or components of cosmetics [3,4,5]. Many phytopharmaceuticals exhibit anticancer, anti-inflammatory, or antibacterial effects [6,7,8]. The commonly known plant-delivered drugs are morphine, paclitaxel, quinine, or vinblastine [9,10,11,12]. Despite its great importance, it’s hard for industry to produce vast amounts of plant secondary metabolites due to their complex chemical structure, chirality, and low concentrations in harvested plant biomass [13,14]. Currently, the in vitro cultures of plant biomass have become an efficient method for the renewable production of plant secondary metabolites. The in vitro systems allow for maintaining the plant biomass cultures under a sterile and controlled environment independently of weather conditions [15]. The in vitro systems facilitate the application of genetic engineering methods to regulate the plant secondary metabolite pathways to enhance the production of desired bioactive compounds [16].

Transgenic roots can be efficiently cultured in bioreactors varying in working volume, i.e., from milliliters up to cubic meters of culture media. Numerous advantages characterize transgenic root cultures, such as biomass suitable for in vitro systems and easy to scale up. The transgenic roots can grow in a hormone-free medium and produce secondary metabolites that generally do not occur in the parent plant [17]. The primary interest in using transgenic roots is maximizing the production of bioactive secondary metabolites. *Rindera graeca* transgenic roots are known for their application in obtaining the naphthoquinones, i.e., rinderol, a potential inducer of apoptosis in cancer cells [18]. Using biocompatible, high porosity materials for biomass immobilization enhances nutrient and oxygen transfer, promotes root proliferation, and supports extracellular secretion of biosynthesized bioactive compounds [19]. Applying proper hydro- or lipophilic materials that additionally allow for in situ extraction of secondary metabolites, by their adsorption directly from the culture system, led to an increase in the yield of these biomolecule biosynthesis [20]. The other technique of improving the production of secondary metabolites and biomass growth is elicitation, which involves using biochemical stress agents to stimulate the plant defense mechanism to achieve higher productivity of bioactive compounds and/or enhance biomass proliferation. The elicitors belong to the broad spectrum of biotic and abiotic substances with exogenous or endogenous origin. The chitosan, methyl jasmonate, temperature, or UV can be applied to elicit plant biomass [21]. Chitosan is an elicitor commonly known for its low toxicity and high biocompatibility [22,23]. This polysaccharide natively occurs in the cell wall of fungal pathogens or shells of invertebrates, and that’s why it is a natural inducer of plant defense mechanisms and, consequently, allows for increased production of secondary metabolites [24].

The scaffolds are widely used in cell cultures in tissue engineering and recovery. The application of the scaffolds in tissue engineering allows for the reconstruction of the architecture and functionality of the native biosystems. They support the cellular processes like adhesion, proliferation, differentiation, and cell migration, which are necessary for proper tissue regeneration. To ensure favorable conditions for nutrient supply, gas exchange, and cell migration, the scaffolds should be porous and biocompatible with the culture tissue [25]. The application of chitosan, which is a biocompatible, non-toxic biomaterial, can be beneficial for cultured biomass. Chitosan can be applied in tissue engineering as nanoparticles, hydrogels, or membranes, and its main application is to support the growth of cells [26]. Furthermore, the chitosan can be integrated into biopolymers, like polylactide, to obtain scaffolds with better mechanical properties. Such materials provide suitable conditions for high cell viability cultures (>95%) and are a promising tool in tissue engineering [27].

In this study, we hypothesized that the incorporation of chitosan into PLA scaffolds would enhance the metabolic activity of *R. graeca* transgenic roots, due to the bioactive nature of chitosan, and the physicochemical properties of chitosan, i.e., its origin, molecular mass, and concentration, would have a significant influence on biomass proliferation and rinderol biosynthesis. We quantitatively investigated the effect of hybrid poly(lactic)–chitosan (PLA–chitosan) scaffolds on in vitro biomass proliferation and rinderol production in *R. graeca* hairy roots. Effects of chitosan origin (i.e., squid or fungal), molecular mass (in the scope of 350–1800 kDa), and concentration (up to 45%) in the developed hybrid material have been quantitatively identified, and the results were compared to the reference system containing an unmodified PLA-based scaffold. The new proposed system innovatively combines three bioengineering techniques: (i) the in situ elicitation of biomass by chitosan molecule residue in the scaffold structure, (ii) the plant biomass immobilization on the scaffold made of hybrid PLA-chitosan biomaterial, and (iii) the in situ extraction of secondary metabolites.

The primary goal of this study was to achieve a significant enhancement in the specific yield of rinderol production. To understand the mechanisms driving this intensification, we formulated competing hypotheses regarding the interplay of our three integrated bioengineering strategies:Elicitation: We hypothesized that chitosan elicitation would be the dominant mechanism, directly stimulating the rinderol biosynthetic pathway and serving as the primary driver of increased specific yield.Sorption: We considered in situ sorption a critical enabling mechanism, which would primarily affect the total yield by alleviating product-related feedback inhibition.Immobilization: We hypothesized that immobilization would play a supportive role by improving mass transfer and maintaining a healthy, metabolically active biomass capable of efficiently responding to the elicitation stimulus.

To test these hypotheses, we quantitatively investigated the effect of the hybrid poly(lactic)–chitosan (PLA–chitosan) scaffolds on in vitro biomass proliferation and rinderol production in *R. graeca* transgenic roots.

## 2. Results

### 2.1. Hybrid PLA-Chitosan Scaffolds Characterization

The shape of the PLA constructs used in this study is presented in Figure 1A,B. All constructs were prepared as flat cylindrical platforms with a 4 cm diameter and ca. 0.5–0.7 cm height, which provided a solid surface for *R. graeca* hairy root adherence. After inoculation, the PLA scaffolds lift transgenic root biomass over the surface of the medium, as shown in Figure 1C. It protects the fragile biomass of hairy roots from mechanical and hydrodynamic stress and enhances the accessibility of oxygen for cultured biomass.

The SEM images of three PLA constructs contain: (i) unmodified PLA, (ii) PLA modified with fungal chitosan, and (iii) PLA modified with squid chitosan are presented in Figure 2. The internal structure of all scaffolds used in the study was highly developed. There were many pores with irregular shapes and sizes (from around 200 µm to around 3000 µm). The medium size of the walls varies from 200 µm to 1000 µm, and on each of them occur small cavities, which increase the total surface of the material. The surface of the constructs modified with chitosan (Figure 2B,C) was sharper than in the unmodified PLA scaffold (Figure 2A).

The concentrations of the chitosan in all applied constructs are presented in Table 1. The highest concentration was observed in the construct modified with a 5% solution of chitosan, characterized by a 350 kDa (%*C_s_* = 44.75%), but the lowest concentration was observed for the scaffold modified with 1% fungal chitosan solution, characterized by a 1800 kDa (%*C_s_* = 18.25%). No significant difference in the chitosan concentration between fungal and squid chitosan was observed when the same solution concentration was applied. The increase of chitosan concentration in the preparation solution used for modification directly resulted in increased chitosan concentration in constructs. However, the chitosan concentration in modified constructs decreased along with increased chitosan particle size.

All the scaffolds had open pores, which relates to high values of the mass absorbability (*A_m_*) presented in Figure 3. The highest value of the *A_m_* was achieved in the culture system with the scaffold made of unmodified PLA (*A_m_* = 12.84 g g^−1^). The lowest value of the *A_m_* was observed for the scaffold made of the PLA modified by 2% solution of the fungal chitosan characterized by 1500 kDa (*A_m_* = 5.09 g g^−1^). There is no clear correlation between the increase in the chitosan origin, concentration, or molecular mass and the *A_m_* of the construct. In general, adding chitosan led to a decrease in the *A_m_* of the constructs.

The values of the contact angle obtained for all studied PLA scaffolds are presented in Figure 4. The lowest value of contact angle (99.64°) was observed for the unmodified PLA scaffold, and the highest value (119.83°) was noticed for the PLA construct modified with 5% fungal chitosan solution characterized by 350 kDa. The differences between all materials weren’t significant, but in general, adding chitosan causes an increase in the contact angle values. Additionally, a slight increase in contact angle values was observed with increased chitosan concentration in scaffolds. However, a slight decrease in the contact angle value was noticed with the increased chitosan molecule size.

### 2.2. The Impact of the PLA Scaffold

The values of dry biomass increase (*DB*), specific growth rate (*µ*), and the yield of the rinderol production (*Y_P/X_*), for the culture system with and without an unmodified PLA scaffold, are presented in Figure 5. The values of *DB* and *µ* for both systems are on comparable levels. In the culture system with an unmodified PLA scaffold, *DB* and *µ* values reached 3.80 [-] and 1.95 × 10^−3^ [h^−1^], respectively, and in the culture system without a scaffold, 3.89 [-] and 2.07·× 10^−3^ [h^−1^], respectively. In the case of the *Y_P/X_* values, the yield of rinderol production reached 5.76 [µg g_DW_^−1^] in the culture system with an unmodified PLA scaffold, and for the culture system without polymeric support, the rinderol concentration does not reach the detection level.

### 2.3. The Screening of the Chitosan Origin

The values of *DB*, *µ*, and *Y_P/X_* achieved from the *R. graeca* hairy roots cultured on PLA scaffolds modified with squid and fungal chitosan, compared to the culture system with an unmodified PLA scaffold, are presented in Figure 6. The highest value of DB was observed for the culture system that contains the unmodified PLA scaffold (*DB* = 3.80 [-]). The slightly lower value of *DB* was noticed for the culture system with PLA scaffold modified by squid chitosan (*DB* = 3.49 [-]). The culture system supported by the PLA scaffold modified with fungal chitosan was characterized by the lowest *DB* value (*DB* = 2.35 [-]). In the case of *µ* value, the highest value of this parameter was observed for the culture system containing the unmodified PLA scaffold (*µ* = 1.95·× 10^−3^ [h^−1^]). The significantly lower values of *µ* were noticed for both culture systems supported by PLA scaffold modified with squid and fungal chitosan (*µ* = 1.25·× 10^−3^ [h^−1^], and *µ* = 1.20·× 10^−3^ [h^−1^], respectively). The highest *Y_P/X_* value was obtained for the culture system containing PLA scaffold modified with fungal chitosan (*Y_P/X_* = 232.56 [µg g_DW_^−1^]. It was around 6 times more than in the culture system containing PLA scaffold modified with squid chitosan *Y_P/X_* = 38.44 [µg g_DW_^−1^]) and around 40 times more than in the control culture system with unmodified PLA scaffold (*Y_P/X_* = 5.76 [µg g_DW_^−1^]).

### 2.4. The Effect of Chitosan Molecular Mass on Transgenic Root Cultures

The *DB*, *µ*, and *Y_P/X_* values obtained for cultures that contain PLA scaffolds with different molecular mass fungal chitosan are presented in Figure 7. The increase in *DB* values was observed with the increase in molecular mas of the chitosan, which was used for surface modification of the PLA scaffold. The highest value was obtained for the culture with PLA scaffold modified with 1800 kDa fungal chitosan (*DB* = 3.38 [-]). Noticeably lower value was observed for the PLA scaffolds modified with the 1500 kDa fungal chitosan (*DB* = 2.88 [-]). The lowest *DB* value was observed in the culture system that contained PLA scaffolds modified with 350 kDa fungal chitosan (*DB* = 2.35 [-]). The *µ* values also increased with the chitosan molecular mass applied to modify the PLA scaffold. The highest value of *µ* was obtained in the culture with PLA scaffold modified with 1800 kDa fungal chitosan (*µ* = 1.72·× 10^−3^ [h^−1^]). A noticeably lower *µ* value was observed for the culture systems containing PLA scaffold modified with 1500 kDa and with 350 kDa fungal chitosan (*µ* = 1.30·× 10^−3^ [h^−1^], and *µ* = 1.20·× 10^−3^ [h^−1^], respectively). However, the highest *Y_P/X_* value was obtained for the culture system containing PLA scaffold modified with 350 kDa fungal chitosan (*Y_P/X_* = 232.56 [µg g_DW_^−1^]). Significantly lower values, about 12 and 16 times lower, were observed for the cultures supported by PLA scaffold modified with 1500 kDa and with 1800 kDa fungal chitosan (*Y_P/X_* = 19.38 [µg g_DW_^−1^], and *Y_P/X_* = 14.63 [µg g_DW_^−1^], respectively).

### 2.5. The Impact of Chitosan Concentration on R. graeca Hairy Root Cultures

The values of the *DB*, *µ*, and *Y_P/X_* obtained in cultures containing PLA scaffolds modified with 2%, 3%, and 5% concentrations of the fungal chitosan solution are presented in Figure 8. The *DB* values increase with the increased chitosan concentration in the modified PLA scaffolds. In the cultures with the PLA scaffolds modified with 2%, 3%, and 5% fungal chitosan solution, the following values of *DB* were observed: 2.35 [-], 2.82 [-], 3.11 [-], respectively. The highest value of *µ* was observed in the culture system containing PLA scaffolds modified with 5% chitosan solution (*µ* = 1.54·× 10^−3^ [h^−1^]). A similar value of *µ* was noticed for the culture system with PLA scaffolds modified with 3% chitosan solution (*µ* = 1.52·× 10^−3^ [h^−1^]). A slightly lower *µ* value was observed for the culture system containing PLA scaffolds modified with 2% chitosan solution (*µ* = 1.21·× 10^−3^ [h^−1^]). Similarly to the *DB* value, the highest *µ* value was obtained in the culture system with PLA scaffolds modified with 5% chitosan solution. Still, it is not significantly higher than the *µ* value for the culture system with PLA scaffolds modified with 3% chitosan solution. The *Y_P/X_* reached the highest value for the culture system containing PLA scaffolds modified with 3% chitosan solution (*Y_P/X_* = 3660.47 [µg g_DW_^−1^]). The value of this parameter obtained for the other cultures was noticeably lower. For the culture system with PLA scaffolds modified with 5% chitosan solution, the *Y_P/X_* value was around 2.5 times lower than the *Y_P/X_* value observed for the best culture system (*Y_P/X_* = 1543.52 [µg g_DW_^−1^]). In the case of the culture system containing PLA scaffolds modified with 2% chitosan solution, the *Y_P/X_* value was around 16 times lower than the highest *Y_P/X_* value noticed in this study (*Y_P/X_* = 232.56 [µg g_DW_^−1^]).

## 3. Discussion

In the literature, *R. graeca* transgenic roots are commonly approved biomass producing naphthoquinones—organic compounds structurally related to naphthalene [28,29,30]. They are used as bioactive compounds of drugs or dyes (e.g., in miniaturized photo-sensible sensors) [31,32,33]. In this study, applying the PLA-chitosan scaffold for cultures of *R. graeca* transgenic roots allows for the efficient production of rinderol, i.e., 2-methoxy-5O, 6-(isohex-1-ene-1,2-diyl)-5,8-dihydroxynaphthalene-1,4-dione. This plant secondary metabolite belongs to the naphthoquinones, and its potential anti-cancer properties are currently being studied [18].

In our study, three different bioengineering methods, i.e., elicitation [34], immobilization [19], and in situ extraction [20], have been applied to intensify the rinderol production in *R. graeca* transgenic roots. The PLA scaffolds were previously applied as a platform for biomass immobilization and as a material for lipophilic rinderol in situ extraction, and chitosan was used as a potential elicitor [18,22]. Based on the obtained results, the polymeric scaffolds can be described as porous and lipophilic materials based on the obtained results. The evidence of lipophilicity is the values of the contact angle higher than 90° obtained for all variants of the used constructs [35]. Another confirmation of scaffolds’ lipophilicity is the high values of mass absorbability obtained from olive oil. The decrease in mass absorbability for PLA scaffolds modified by chitosan resulted from partial filling of scaffold pores by chitosan during surface modification. However, the chitosan addition slightly increased the lipophilicity of the material (the increase in contact angle values has been observed), which makes better conditions for the sorption of the rinderol. Both parameters, high contact angle values and high mass absorbability, show the high potential of the proposed material for effective in situ extraction of the lipophilic rinderol. The high porosity of PLA-chitosan scaffolds is confirmed by SEM pictures (Figure 2). Highly developed surface of the scaffolds with open pores provides perfect conditions for adhesion of *R. graeca* hairy roots to polymeric material, as shown in Figure 9. *R. graeca* hairy roots outgrow the PLA scaffolds, forming a homogenous unit, even leading to destruction of the constructs (Figure 9B). This phenomenon makes it impossible to separate the fresh biomass from the polymeric scaffold. Separating hairy roots from the scaffolds was only possible after lyophilization of biomass and scaffolds together. However, after lyophilization, separating hairy roots from the scaffolds was difficult because numerous small particles of the scaffold were attached to the biomass of tangled roots. The impossibility of hairy roots separation from the scaffolds also made it impossible to determine the fresh biomass increase, which is why we only based the results of biomass proliferation on the *DB* value.

The content of rinderol in a particular fraction of each culture system is presented in Figure 10. In almost every culture system, the amount of rinderol found in the scaffold fraction was significantly higher than in the culture medium. The exception was the culture system containing PLA scaffold modified with squid chitosan, but the mass of the obtained rinderol was very low. As we can see, the polymeric scaffold sorbed it effectively when the *R. graeca* hairy roots secreted the rinderol into the culture medium. The contents of rinderol in scaffold and root fractions in all tested PLA-chitosan scaffolds varied due to the difficulties in separating hairy roots from the scaffolds. Small parts of the scaffold attached to the roots were extracted with the root fraction, which probably increased the content of rinderol in the root fraction.

Analyzing the data presented in Figure 10, we can conclude that the hybrid PLA-chitosan scaffolds are an effective material for in situ extraction of lipophilic rinderol because most of the rinderol secreted from the roots was absorbed by the scaffolds. However, we cannot conclude that in situ extraction was the main reason for the increased rinderol production. The mass of rinderol produced in the tested culture systems supported with PLA-chitosan scaffolds differed significantly. However, all PLA-chitosan scaffolds had comparable contact angles and *A_m_* values, which indicates similar properties for in situ extraction. Due to that, we hypothesize that chitosan properties were the main factor influencing the rinderol production.

Applying chitosan for the surface modification of the PLA scaffolds stimulates the production and secretion of rinderol by *R. graeca* transgenic roots biomass. The elicitation effect of the chitosan is known and confirmed [22,23,36,37]. For this study, squid and fungal chitosan were used. Fungal chitosan led to a 6 times greater value of *Y_P/X_* than the same amount of squid chitosan. Structural differences between chitosan, the product of chitin deacetylation, caused this. Fungal chitin is characterized by the α form; however, the β form is characterized by squid-originated chitosan. Both structural forms differ based on the hydrogen bond between the hydroxyl and amide groups, and consequently, it affects the stability, reactivity, and biopolymer conformation. Next, both polysaccharides have various levels of acetylation. Fungal chitosan acetylation level varies between 10 and 20%, while squid chitosan has around a 50% acetylation level. Additionally, fungal chitosan is characterized by shorter molecules than squid chitosan [38,39]. There are also differences in *DB* and *µ* parameters, stemming from structural differences. After adding the chitosan, the values of *DB* and *µ* decreased, and the application of fungal chitosan resulted in a significant drop in these values. Because of the higher impact on the production of the fungal chitosan, it was picked for further investigations. Its ability to stimulate the production of rinderol could be the outcome of its abilities to activate the proper gene expression that play a crucial role in the biosynthesis of plant secondary metabolites. Fungal chitosan is a biotic elicitor that mimics pathogen attack and activates plant defense response [40]. It could cause the oxidative burst, which activates the mitogen-activated protein kinase signaling (MAPK). MAPK is related to the expression of genes specific to the secondary metabolites biosynthetic pathway among various plant species [22].

The next step was to explore the influence of the molecular mass of the fungal chitosan on the proliferation of hairy roots and production of the rinderol. Chitosan molecular mass directly corresponds to the viscosity of the chitosan molecule. With the increased molecular mass, viscosity also increases, and, as a consequence, physical and chemical properties change, resulting in a change in biological activity [41]. According to the results presented in Figure 7 and Table 1, it can be concluded that with the increased molecular mass of the used chitosan, the impact on the production of secondary metabolites is decreased. However, the effect on plant biomass proliferation is opposite. It is essential to mention that the increased molecular mass of the used chitosan led to a decrease in the final chitosan concentration in the PLA-chitosan scaffolds, which correlates with its molecular size. It is more difficult for bigger compounds to absorb onto the tested scaffolds. It could be an additional reason for the obtained results of biomass proliferation and rinderol productivity.

Analyzing the results of the influence of different amounts of chitosan on the modified constructs, we can conclude that an increase in the amount of chitosan causes an increase in dry biomass. An analogical effect was observed in the cell suspension culture of the *Eurycoma longifolia*, where the proliferation of the biomass was increased along with the increase of the chitosan concentration, which caused decreased biomass proliferation [42]. We hypothesize that further growth of the chitosan concentration in our study could give the same effect. However, considering the results describing the productivity of *R. graeca* hairy roots, it shows that productivity increases with increasing chitosan concentration. After reaching a maximum value in this study, productivity decreases. Considering the data from Table 2 and the results obtained, it can be concluded that the culture system tested during this study is very effective regarding the naphthoquinones production in the *R. graeca* transgenic roots. The productivity characterizes the developed hybrid PLA-chitosan scaffolds at a comparable level to the highest productivity found in the literature, i.e., biomass immobilized on a polyurethane foam raft [18]. Crucially, both the PLA–chitosan scaffold and the polyurethane foam raft induce a metabolic response in the transgenic root biomass; however, the mechanism differs: the PLA–chitosan scaffold achieves this through the presence of chitosan elicitor, while the polyurethane foam induces the response via heavy metal contaminants it contains [18]. The significant advantage of the PLA–chitosan scaffold is that its composition is fully known, unlike the polyurethane foam, which grants us greater control and reproducibility over time. In comparison to the PLA-chitosan scaffold, the MTMS xerogel only serves an in situ extraction function in the system, which is reflected in the lower *Y_P/X_* values (Table 2) [19].

We hypothesize that the combination of in situ extraction, immobilization, and elicitation used at the same time can be beneficial. Such methods used individually do not demonstrate such significant potential for productivity stimulation. However, manipulating plant growth regulators achieves good results with a simpler culture system for naphthoquinones production in *Impatiens balsamina* biomass [43]. In the future, effective bioengineering methods should be combined with process scale-up and genetic engineering methods to maximize the productivity of in vitro culture systems. However, from a practical perspective, a key issue is the impossibility of efficiently separating the root biomass from the scaffolds after cultivation, which is only partially possible after lyophilization and complicates subsequent product extraction steps. This negatively affects both the process reproducibility in terms of product recovery and the potential automation of the harvesting stage on an industrial scale. These drawbacks are particularly significant when compared to established systems like suspension or hairy root cultures in bioreactors, which allow for simple biomass recovery via filtration or centrifugation and completely bypass the high costs associated with scaffold fabrication. Regarding cost-effectiveness, the multi-step and time-consuming scaffold preparation procedure, which includes multi-day cooling, washing, and drying processes, the application of the proposed system may generate high operational costs, posing a barrier to economical large-scale production. Although the current research confirms high biochemical efficacy, the authors state that the research is limited to the laboratory scale and needs validation in larger-scale systems. Future work should therefore focus on simplifying the scaffold manufacturing technology and developing methods for efficient biomass harvesting, which is essential to transform this promising method into a reproducible and cost-effective industrial process.

**Table 2 ijms-26-10668-t002:** The comparison of the previously published quantitative data on *Y_P/X_* characterizing the naphthoquinone-originated bioproducts produced in various systems for in vitro bioprocessing of transgenic roots of *Boraginaceae* representatives.

Transgenic Root Origin	Produced Plant Secondary Metabolites	In Vitro Culture System	*Y_P/X_* [µg g_DW_^−1^]	Reference
*R. graeca*	naphthoquinones	biomass immobilization and in situ extraction of metabolites on MTMS xerogel	632	[19]
		biomass immobilization and in situ extraction of metabolites on polyurethane foam	652	[19]
	deoxyshikonin and rinderol	biomass immobilization and in situ extraction of metabolites on xerogel functionalized by methyl groups	229	[44]
	deoxyshikonin	bioprocess scaling-up performed in a single-use wave-mixed bioreactor	165	[45]
	rinderol	biomass immobilization and metabolites in situ extraction on polyurethane foam raft	3770	[18]
		simultaneous biomass immobilization, elicitation, and in situ extraction of metabolites, performed with hybrid PLA-chitosan scaffolds	3660	current study
*Lithospermum canescens*	acetyloshikonin	screening plant growth regulators and culture medium	2720	[46]
*Impatiens balsamina L.*	naphthoquinones	screening plant growth regulators	2970	[46]

Considering all the results obtained during this study, it can be concluded that chitosan and its properties were the leading cause of the increased production of the rinderol in the *R. graeca* hairy roots culture systems. The results of our study indicate that the fungal chitosan with low acetylation level and low molecular mass resulted in higher productivity of rinderol. Similar conclusions were made by Coelho and Romano, who concluded that the small molecules and low acetylation level of chitosan can result in its higher biological activity [47]. This effect is probably caused by the structure of chitosan and the biological activity, which makes it the closest to fungal plant pathogens. Consequently, hairy roots could recognize chitosan as a fungal infection of the plant, and their response can be stronger.

## 4. Materials and Methods

### 4.1. Hybrid PLA-Chitosan Scaffolds Preparation and Characterization

Firstly, the 3% *w*/*v* poly(lactic acid) (PLA) (Nature Works, Minneapolis, MN, USA) solution in 1,4-dioxane (POCH) was prepared and left for continuous stirring for 24 h at room temperature. Next, in the 55 °C water bath, 75 µL of Milli-Q water per 1 mL of 1–4 dioxane was added to the PLA solution. Then, the solution was left to stir for the next 24 h at room temperature.

After that, about 75 mL of previously prepared PLA solution was poured into the rounded forms. Then, forms filled with the solution were put in the refrigerator (−18 °C) for 24 h. Next, scaffolds were removed from the forms and placed in the Erlenmeyer flask with 300 mL of chilled methanol (POCH). Then, each flask with a scaffold was put in the refrigerator (−18 °C) for 5 days. After this time, every scaffold was placed in distilled water for washing for 24 h. Water was exchanged twice, after 3 and 6 h. PLA scaffolds were dried at 45 °C for 72 h.

For scaffold surface modification, aqueous solutions of chitosan acetate were used. Four types of chitosan differing in molecular weight and biological source were employed. A low molecular weight squid-derived chitosan (viscosity: 10–100 cps, average molecular weight 580 kDa, Pol-Aura, Zawroty, Poland, cat. no. PA-03-4907-E) and three fungal-derived chitosans: a low molecular weight grade (viscosity: 10–120 cps, average molecular weight 350 kDa, Pol-Aura, Zawroty, Poland, cat. no. PA-03-1067-U), a high molecular weight grade (viscosity: 100–300 cps, average molecular weight 1500 kDa, Pol-Aura, Zawroty, Poland, cat. no. PA-03-5053-P), and a very high molecular weight grade (viscosity: 2000–3500 cps, average molecular weight 1800 kDa, Pol-Aura, Zawroty, Poland, cat. no. PA-03-8689-P) were used.

According to the manufacturer’s data sheets, all chitosans used met microbiological quality standards, with yeast and mold counts ≤ 100 CFU g^−1^, and additionally for squid-derived chitosan, with bacteria of *Escherichia coli* ≤ 3 CFU g^−1^. Heavy metal contamination levels did not exceed the following limits: arsenic (As) ≤ 1 mg kg^−1^, lead (Pb) ≤ 0.5 mg kg^−1^, mercury (Hg) ≤ 0.1 mg kg^−1^, and cadmium (Cd) ≤ 1 mg kg^−1^. The essential structural and quality parameters (viscosity, average molecular weight, origin, microbiological purity, and heavy metal content) were obtained from the manufacturer’s technical documentation available on the supplier’s website, as these characteristics are known to influence both the physicochemical properties and biological behavior of chitosan.

Water solutions of chitosan acetate were prepared as follows. Firstly, a 1–5% *w*/*w* water acetic acid (POCH, Warsaw, Poland) solution was prepared in the Erlenmeyer flask. Then, a conical flask was put in the 60 °C water bath, where the proper chitosan was added until it was completely dissolved. After that, the solution was stirred for 24 h at room temperature. The final solution concentrations depended on the viscosity of the used chitosan and the targeted chitosan mass content of the PLA scaffolds’ surface.

The PLA scaffolds were singly put in the container. Next, every scaffold was flooded with 20 g of the proper chitosan acetate solution and put in the vacuum chamber (10 mbar, 30 min) for saturation. Then, PLA scaffolds were wrapped with parafilm and were placed in the refrigerator (−18 °C) for 24 h on the Petri dish. After this time, scaffolds with modified surfaces were dried in the vacuum chamber (10 mbar, 48 h) without heating.

The next step was preparing the sodium bicarbonate water-methanol (3:1 *v*/*v*) solution with 1.12–5.60% *w*/*v*. Once again, scaffolds were singly put in the container and flooded with 30 mL of sodium bicarbonate solution. Then, the container was placed in the vacuum chamber (10 mbar, 15 min) for saturation. Next, saturated scaffolds were put in the distilled water for washing for 3 h. Water was exchanged twice, after 1 and 2 h. After that, the PLA with a modified surface was dried at 45 °C for 72 h.

The effect of chitosan surface modification was determined quantitatively by mass measurement before and after the process.

SEM imaging (TM400, Hitachi, Tokyo, Japan, and MAIA 3, TESCAN, Brno, Czech Republic) was conducted for structural analysis of the scaffolds. Small sections of the outer and inner mass of the scaffolds were cut out. Samples were coated with a 4 nm thick platinum layer beforehand.

The concentration of the chitosan in PLA scaffolds (%*C_s_*)was determined with the following formula:%*C_s_* = ((*m_cs_* − *m*_0_)/*m*_0_) · 100%,(1)
where *m_cs_* is the mass of the material after the chitosan modification, and *m*_0_ is the mass of the material before the modification.

For the mass absorbability (*A_m_*) calculation, samples were placed in olive oil (density equals 0.915 g cm^−3^ at 20 °C) for 24 h. The procedure was carried out thrice for each material. Then, mass absorbability was calculated according to the following formula:*A_m_* = ((*m_wet_* − *m_dry_*)/*m_dry_*) · 100%,(2)
where *m_wet_* is the mass of the material after soaking in olive oil for 24 h, and *m_dry_* is the mass of the dry material.

The water in air contact angle was measured with a sessile drop method using the goniometer (OCA 25, DataPhysics Instruments, Filderstadt, Germany). The contact angle was determined using 10 μL droplets, and each droplet was measured five times. Each of the measurements was done at 24 °C.

### 4.2. Inoculum of Rindera graeca Transgenic Roots

The KT17 line of the *Rindera graeca* transgenic roots used in this study was originally developed at the Faculty of Pharmacy of the Medical University of Warsaw [48]. The inoculum of hairy roots was cultured in 50 mL of DCR hormone-free medium (PhytoTech Labs, Lenexa, KS, USA) in a 250 mL Erlenmeyer flask on an oscillatory shaker (ISS-9100, Lab Companion, Billerica, MA, USA) at 105 rpm at 24 °C in the dark. Culture was maintained for 28 days, and after this time, the passage of the biomass was performed in sterile conditions.

### 4.3. Bioprocessing of Rindera graeca Transgenic Roots on PLA-Chitosan Scaffolds

At first, presterilized PLA-chitosan scaffolds were added independently to the 250 mL Erlenmeyer flask with 50 mL of DCR hormone-free medium (PhytoTech Labs, Lenexa, KS, USA), in sterile conditions. Then, 1 g of the inoculum, 28-day *R. graeca* transgenic roots, was placed on the top of each scaffold, which was floating on medium. Next, every culture system was incubated for 28 days in darkness on an oscillatory shaker (ISS-9100, Lab Companion, Billerica, MA, USA) at 105 rpm at 24 °C. The culture system with biomass immobilized on the non-modified PLA scaffold was a control system.

After 28 days of culturing, the *R. graeca* hairy roots were harvested, and the fresh biomass mass was measured. Additionally, scaffolds, the hairy roots, and filtered medium were collected and put in the refrigerator at −20 °C. Next, every solid-state refrigerated probe was lyophilized at −20 °C for 5 days (ALPHA 1-4 LSC, Christ, Memmingen, Germany). After that, transgenic roots were ground in the mortar. Prepared, filtered medium and hairy roots were extracted with n-hexane (Merck, Darmstadt, Germany). The scaffolds were extracted with methanol (Merck, Darmstadt, Germany). Every sample was sonicated (BK-9050, APTEL, Białystok, Poland) until the solvent lost color.

### 4.4. Phytochemical Analysis of Extracts

Phytochemical analysis was carried out using chromatographic methods. For this purpose, the HPLC system (DIONEX 3000, Thermo Scientific, Waltham, MA, USA) connected with an automated sample injector (ASI-100, Thermo Scientific, Waltham, MA, USA) and UV-Vis diode-array detector (UVD 340S, Thermo Scientific, Waltham, MA, USA) was used. The procedure was carried out under the following conditions: flow rate 1.5 mL/min, gradient elution—acetonitrile (60–80%) with 0.04 M ortho-phosphoric acid (40–20%), injection volume 20 mL, duration time 15 min, EC Nucleosil 120-7 ODS packed column (250 × 4.6 mm, 7 μm particle diameter, 120 Å pores) (Macherey-Nagel, Düren, DE, USA).

Eluent absorbance was observed at 215 nm, 237 nm, 350 nm, and 436 nm wavelengths. Naphthoquinone derivatives were identified with standard applications. The Limit of Detection (LOD) for rinderol by HPLC analysis was 0.1 µg per 1 mL of extract.

### 4.5. Mathematical Methods

The amount of dry biomass (*DB*) characterizes the proliferation of transgenic roots after 28 days of culturing. *DB* was calculated with the following equation:*DB* = *m_DB28d_*/*m*_*DB*0*d*_ [-],(3)
where *m_DB28d_* is the dry biomass weight of transgenic roots after 28 days, and *m_DB_*_0*d*_ is the dry biomass weight of the hairy roots inoculum.

The values of the specific growth rate (*µ*), which describe the growth rate of the *R. graeca* hairy roots in the culture systems, were determined with the following equation:*µ =* (ln(*m_DB28d_*) − ln(*m_DB_*_0*d*_))/*t* [h^−1^],(4)
where *DB* is the dry biomass, and *t* is the time of the culture.

The yield of the rinderol production per amount of the dry biomass (*Y_P/X_*) was calculated with the following equation:*Y_P/X_* = *m_n_*/*m_DB28d_* [µg g_DW_^−1^],(5)
where *m_n_* is the mass of rinderol obtained in the culture system.

### 4.6. Statistical Analysis

Statistical analysis was performed to identify significant differences among the experimental groups. Each experimental condition was evaluated in five independent replicates (*n* = 5), where a single culture flask represented the experimental unit. The complete set of experiments was performed once. The normality of data distribution was assessed using the Shapiro-Wilk test, and the homogeneity of variances was evaluated with Bartlett’s test.

A one-way analysis of variance (ANOVA) with Tukey’s HSD post-hoc test was used to compare three or more groups when data met the assumption of equal variances. If variances were not homogeneous, Welch’s ANOVA followed by the Games-Howell post-hoc test was applied. For the comparison of two independent groups, the non-parametric Mann-Whitney U test was utilized. For all statistical tests, a *p*-value of ≤0.05 was considered significant.

## 5. Conclusions

Bioengineering techniques, such as plant biomass in vitro immobilization, biochemical elicitation, and in situ extraction of metabolites, have been combinedly applied for in vitro cultures of *R. graeca* hairy roots, as a biomass producing rinderol, the bioactive naphthoquinone with the potential to induce apoptosis in cancer cells. All related results have been deeply analyzed and comprehensively discussed. The originally prototyped hybrid PLA-chitosan scaffolds fulfilled simultaneously all three functionalities in the studied culture system. Based on the collected data, it has been proven that applying hybrid PLA-chitosan scaffolds to intensify rinderol production was entirely successful. It has also been demonstrated that chitosan was the main factor responsible for increasing the production of rinderol, and the origin of chitosan and its molecular mass and concentration in the scaffold influenced the *Y_P/X_* values reached. Applying chitosan (especially of fungal origin) caused the biomass proliferation to decrease, compared to plain polymeric scaffolds, but increased rinderol production. The influence of chitosan molecular mass, a parameter directly related to the length of chitosan molecules and their molecular mass, was opposite for biomass proliferation and rinderol production, and the increase of chitosan molecular mass resulted in higher biomass proliferation but hurt rinderol production. In the case of the effect of the chitosan concentration in the applied hybrid scaffolds, the increase in the chitosan concentration in the scaffold resulted in higher biomass proliferation and higher production of secondary metabolites. Applying PLA–chitosan scaffold modified with 3% of the fungal chitosan solution characterized by 350 kDa, resulted in up to 635 times higher production of rinderol, compared to applying reference unmodified PLA scaffolds. To finally conclude, the hybrid PLA-chitosan scaffold containing 33% of fungal chitosan, characterized by low molecular mass and acetylation level, may be successfully applied to intensify rinderol production in cultures of *R. graeca* transgenic roots. Despite promising results, the research is limited to the laboratory scale and needs validation in larger-scale systems.

## Figures and Tables

**Figure 1 ijms-26-10668-f001:**
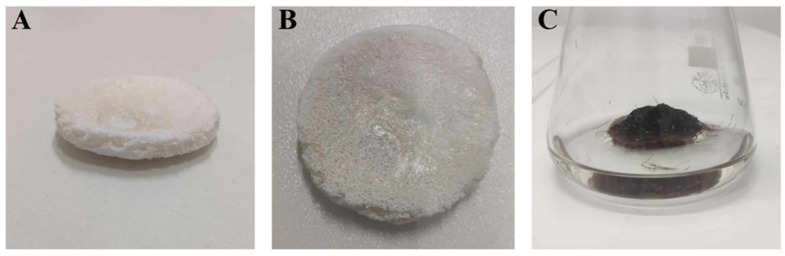
The shape of the PLA constructs used during this study (**A**,**B**), and an exemplary scaffold overgrown by biomass of *R. graeca* transgenic roots (**C**).

**Figure 2 ijms-26-10668-f002:**
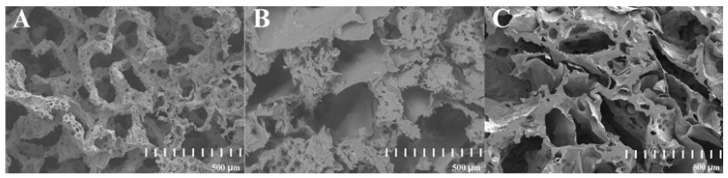
SEM images of the cross-section of three PLA-based constructs: unmodified PLA (**A**), hybrid PLA-chitosan scaffold modified with fungal chitosan (**B**), and hybrid PLA-scaffold modified with squid chitosan (**C**).

**Figure 3 ijms-26-10668-f003:**
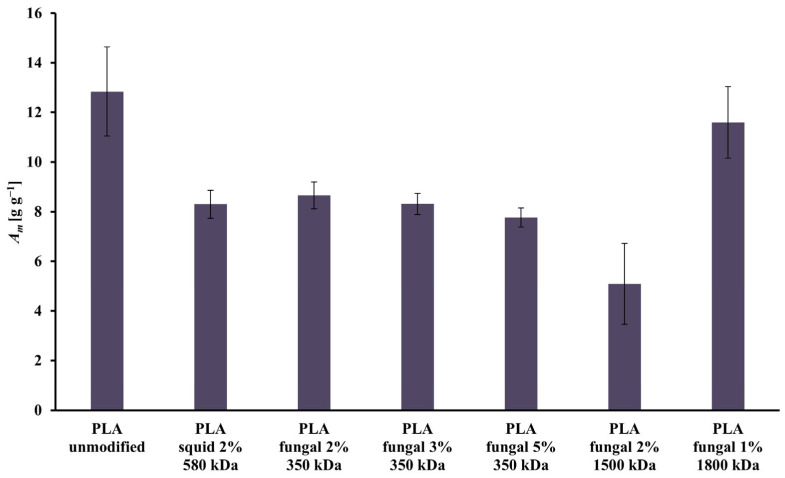
Values of *A_m_* characterizing all scaffolds (i.e., reference unmodified PLA, and hybrid PLA-chitosan scaffolds, varied in chitosan source, concentration, and molecular mass) used for this study.

**Figure 4 ijms-26-10668-f004:**
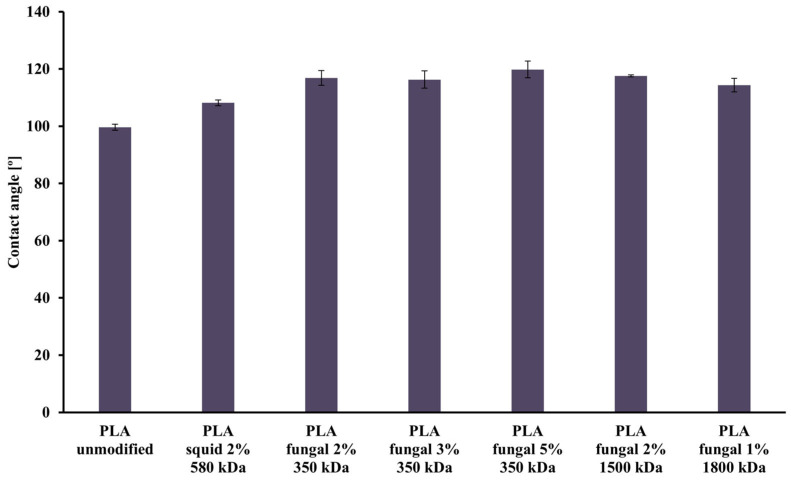
Values of the contact angle characterizing all scaffolds (i.e., reference unmodified PLA, and hybrid PLA-chitosan scaffolds, varied in chitosan source, concentration, and molecular mass) used for this study.

**Figure 5 ijms-26-10668-f005:**
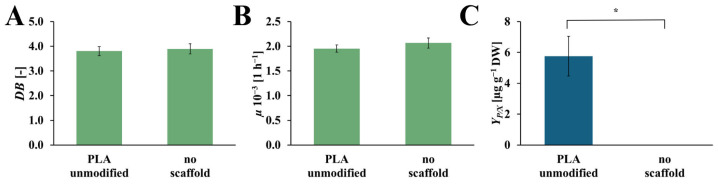
Values of the *DB* (**A**), *µ* (**B**), and *Y_P/X_* (**C**) noted after 28 days of *R. graeca* transgenic root cultures with a referential unmodified PLA scaffold, and without any scaffold incorporated into the culture system. The *p*-values over brackets were determined according to the Mann-Whitney U test. Statistically significant differences at the *p* < 0.05 level are denoted by an asterisk (*).

**Figure 6 ijms-26-10668-f006:**
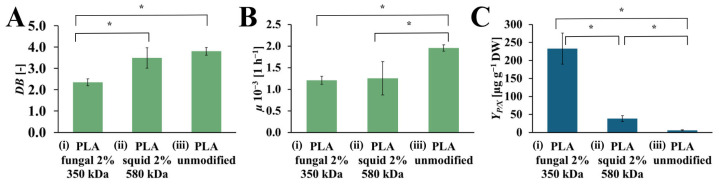
The value of *DB* (**A**), *µ* (**B**), and *Y_P/X_* (**C**) noted after 28 days of *R. graeca* transgenic root cultures supported with: (i) hybrid PLA-chitosan scaffold modified with 2% fungal chitosan solution, characterized by 350 kDa, (ii) hybrid PLA-chitosan scaffold modified with 2% squid chitosan solution, characterized by 580 kDa and (iii) reference unmodified PLA scaffold. A Welch’s ANOVA test was conducted to determine if there were significant differences between the group means. Since the result was significant, post-hoc comparisons using the Games-Howell test were performed to identify which specific groups differed from one another. Statistically significant differences at the *p* < 0.05 level are denoted by an asterisk (*).

**Figure 7 ijms-26-10668-f007:**
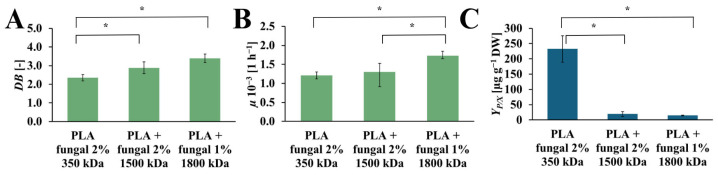
The value of *DB* (**A**), *µ* (**B**), and *Y_P/X_* (**C**) noted after 28 days of *R. graeca* transgenic root cultures supported with hybrid PLA-chitosan scaffolds modified with fungal chitosan characterized by molecular mass of 350 kDa, 1500 kDa, and 1800 kDa. An ANOVA test (**A**,**B**) and Welch’s ANOVA (**C**) test were conducted to determine if there were significant differences between the group means. Since the result was significant, post-hoc comparisons using Tukey’s HSD test (**A**,**B**) and the Games-Howell test (**C**) were performed to identify which specific groups differed from one another. Statistically significant differences at the *p* < 0.05 level are denoted by an asterisk (*).

**Figure 8 ijms-26-10668-f008:**
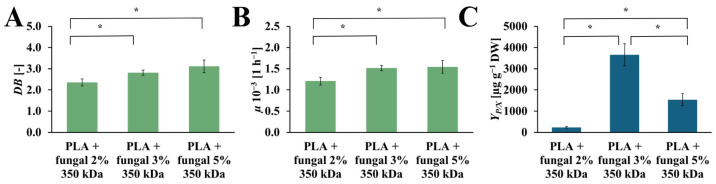
The value of *DB* (**A**), *µ* (**B**), and *Y_P/X_* (**C**) noted after 28 days of *R. graeca* transgenic root cultures supported with hybrid PLA-chitosan scaffolds modified with: 2% fungal chitosan solution, 3% fungal chitosan solution, and 5% fungal chitosan solution, all equally characterized by 350 kDa. An ANOVA test (**A**,**B**) and Welch’s ANOVA (**C**) test were conducted to determine if there were significant differences between the group means. Since the result was significant, post-hoc comparisons using Tukey’s HSD test (**A**,**B**) and the Games-Howell test (**C**) were performed to identify which specific groups differed from one another. Statistically significant differences at the *p* < 0.05 level are denoted by an asterisk (*).

**Figure 9 ijms-26-10668-f009:**
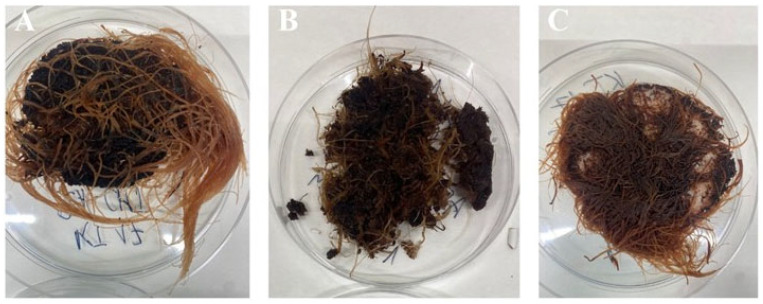
Examples of the *R. graeca* transgenic roots harvested after 28-day cultures supported with hybrid PLA-chitosan scaffolds modified with 2% fungal chitosan, characterized by 350 kDa (**A**), with hybrid PLA-chitosan scaffolds modified with 3% fungal chitosan, characterized by 350 kDa (**B**), and from the reference culture system supported with an unmodified PLA scaffold (**C**).

**Figure 10 ijms-26-10668-f010:**
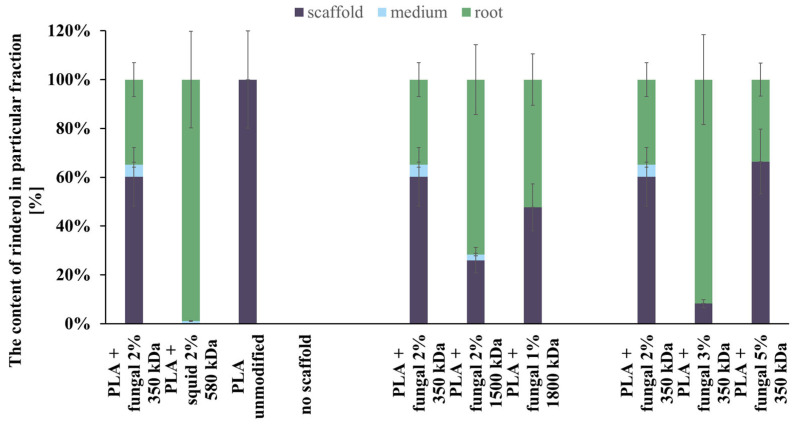
The total content of rinderol chromatographically detected in all fractions of every culture system tested.

**Table 1 ijms-26-10668-t001:** Values of the chitosan concentration characterizing scaffolds used for this study.

Culture System	%C_s_ [%]	
	Value	SD
No scaffold	0.00%	0.00%
PLA unmodified	0.00%	0.00%
PLA + squid 2% 580 kDa	25.46%	1.55%
PLA + fungal 2% 350 kDa	27.20%	1.54%
PLA + fungal 3% 350 kDa	33.15%	2.65%
PLA + fungal 5% 350 kDa	44.75%	8.59%
PLA + fungal 2% 1500 kDa	22.04%	1.60%
PLA + fungal 1% 1800 kDa	18.25%	3.47%

## Data Availability

The raw data supporting the conclusions of this article will be made available by the authors on request.

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
