# Peer review of "Hybrid Poly(Lactic)-Chitosan Scaffold Intensifying In Situ Bioprocessing of Rindera graeca Transgenic Roots for Enhanced Rinderol Production"

_ijms, 2025, doi:10.3390/ijms262110668_

Round 1
Reviewer 1 Report
Comments and Suggestions for Authors
Upon reviewing the manuscript titled "Hybrid poly(lactic)-chitosan scaffold intensifying in situ bioprocessing of Rindera graeca transgenic roots for enhanced rinderol production," the following suggestions are provided:
- It is recommended that the authors prepare a Graphical Abstract to visually summarize the overall workflow and main findings. This would improve the accessibility and visual impact of the article.
- The Introduction provides a good background on Rindera graeca and the role of scaffolds and chitosan in bioprocess intensification. However, it is recommended that the novelty of this study be emphasized more clearly in the final paragraph, with a brief comparison to previous scaffold-based approaches. It is also suggested that the following references be cited, as they provide broader context on chitosan- and plant-derived biomaterials:
- https://doi.org/10.61186/jcc.7.2.2
- http://doi.org/10.3390/jcs9060305
- http://geobioj.com/geo/index.php/geojournal/article/view/9
These references are recommended solely to strengthen the scientific completeness of the Introduction.
- It is recommended that the Conclusion include a brief note on the limitations of the present work (e.g., being limited to laboratory-scale experiments and the need for validation in larger-scale systems).
- A thorough check for grammatical and spelling errors throughout the manuscript is recommended.
Author Response
Dear Reviewer,
Thank you for your thorough and insightful review of our manuscript. We appreciate the time and effort you dedicated to providing these constructive comments, which have significantly contributed to improving the clarity and scientific rigor of our work.
Our detailed responses to each of your remarks are attached below.
Sincerely,
Kamil Wierzchowski

Reviewer 2 Report
Comments and Suggestions for Authors
An inventive and well-organized investigation on the application of hybrid poly(lactic)-chitosan scaffolds to increase rinderol production in transgenic roots of Rindera graeca is presented in the manuscript. An important contribution to plant biotechnology and secondary metabolite engineering is made by the authors' effective integration of immobilization, elicitation, and in situ extraction techniques into a single system. Chitosan characteristics (origin, viscosity, and concentration) and their effects on biomass proliferation and metabolite yield are thoroughly explored in this meticulously designed experiment. For creating scalable culture systems, the discovery that fungal chitosan significantly boosts rinderol yield at the ideal concentration and viscosity is extremely pertinent. Nevertheless, I would want to bring up the following issues for improvements:
- The reason why fungal chitosan increases rinderol production should be explained by the authors using supporting information or research on biological mechanisms (such as defensive gene expression or signaling networks).
- In comparison to current systems, discuss about the practicality, cost-effectiveness, and reproducibility of the potential for industrial application and scale-up.
- Although chitosan elicitation is primarily credited in the discussion for increased rinderol production, the underlying molecular mechanisms (such as altered gene expression or signaling pathways) are still up for debate. The mechanistic explanation would be enhanced by mentioning more focused research or by including at least preliminary data (such as qPCR on defense-related genes).
- The underlying molecular mechanisms (such as signaling cascades or changes in gene expression) are still up for debate, however the discussion primarily credits chitosan elicitation with increased rinderol synthesis. It would enhance the mechanistic explanation to include at least preliminary data (e.g., qPCR on defense-related genes) or to cite more focused studies.
- The current productivity is helpfully compared with literature studies in Table 2, but the debate may be extended to critically examine the reasons why PLA–chitosan scaffolds function similarly or differently from MTMS xerogels and polyurethane foam. It would be easier to understand the practical relevance if you highlighted topics like cost, scalability, or biocompatibility.
- Some figures and tables might be rearranged for easier reading, even though the results are comprehensive. Multiple variables are shown simultaneously in Figures 6–8, for instance; interpretation might be strengthened by the addition of more understandable statistical significance indicators.
Author Response

(The authors gave the same response as above.)

Reviewer 3 Report
Comments and Suggestions for Authors
Most of the measured parameters in the manuscript are related in some way to the concept of viscosity. I did not find a method for measuring viscosity in the manuscript. The viscosity of the samples is sometimes given with a spread of one order of magnitude or more. Clearly, when fabricating porous materials, the viscosity of the material is not the average of the viscosities of the microvolumes occupied by the gas and the known material, so any discussion of viscosity in the manuscript is unnecessary. If the authors could measure the viscosity of the preforms with the required accuracy in real physical units, then we could revisit the manuscript. I don't think this is possible, since the preforms are fabricated with an accuracy of 30% (0.7-0.9 cm).
For scaffolds, the ability to absorb and retain liquids is usually determined by the scaffold's density. Unlike relative units, density is a physical quantity that can be measured very quickly using a scale (with known material density) or using a scale and a graduated cylinder without knowing the density... The authors measured the absorption and retention capacity of liquids using an oil of unknown viscosity. It should be noted that the ability of liquids to penetrate cavities is highly dependent on the viscosity of the oil and can vary by more than one order of magnitude between different oils.
In general, the manuscript is guilty of presenting data not in real units, but in relative units, and even those that are rarely used... For example, the authors express the change in dry mass, which is measured in grams, in DB. Why? The specific growth rate is not in grams per hour, but in some relative units, and even logarithmic ones (although they state it's in h-1).
Overall, the work can be presented as a methodological one. However, the primary research tool used in the manuscript is weighing... Yet, we learn little new about the fundamental aspects of the process. I also believe the work's technological sophistication and its level are not up to par with those of IJMS. I recommend the authors submit their manuscript to the journal Agriculture.
Author Response

(The authors gave the same response as above.)

Reviewer 4 Report
Comments and Suggestions for Authors
- 1. Please list keywords in alphabetical order.
- 2. The goal, using hybrid PLA, chitosan scaffolds to intensify rinderol in Rindera graeca hairy roots, is clearly stated, along with the three “integrated” strategies (immobilization, elicitation, in-situ extraction).
1) However, stage-wise or mechanism-level, a priori predictions are not specified (e.g., whether chitosan origin/viscosity/concentration are expected to affect biomass vs. secretion vs. sorption differentially).
2) Please predefine primary outcomes (e.g., YP/X thresholds) and competing hypotheses (elicitation vs. sorption vs. mass-transfer effects).
- 3. Culture geometry is per-flask scaffold with 1 g inoculum for 28 days.
1) However, the number of independent flasks per condition, whether runs were repeated across days/batches, and how variability enters statistics are not specified.
2) Define the experimental unit (flask), "n" per treatment, and the number of independent runs used in figures/tables.
- 4. Mass absorbability (Am) is measured in olive oil as a proxy for lipophilic uptake; contact angle is measured in water.
1) Please justify olive oil as a surrogate for rinderol partitioning and report whether scaffold extraction recoveries (methanol) vs. root/media extractions (hexane) were cross-validated for bias.
- 5. Fungal chitosan (low viscosity) maximized YP/X while reducing growth (DB, μ).
1) Please report effect sizes with CIs for DB, μ, YP/X and quantify productivity per time and per vessel (e.g., μg·L^-1·d^-1), not only per g DW, to ground the process claim.
- 6. You show that most rinderol is in the scaffold fraction and acknowledge that scaffold fragments adhering to roots likely inflated the “root” fraction.
1) This complicates mass balances and YP/X attribution.
2) Please provide full mass balances with error bars by fraction and a control demonstrating negligible carryover (e.g., dyed scaffold surrogate, or spike-recovery from mixed matrices).
- 7. The no-scaffold system had rinderol at a concentration below the detection limit.
1) Without LOD/LOQ, fold-improvements vs. zero are not interpretable.
2) Report censored-data handling (e.g., substitution at LOD/√2) and show sensitivity of fold-change to plausible LOD values.
- 8. The narrative attributes gains mainly to chitosan elicitation, yet also argues lipophilicity/contact angle and Am support in-situ extraction.
1) Given comparable contact angles and Am across PLA–chitosan variants, the conclusion that elicitation dominates needs direct evidence (e.g., defense-pathway markers, time-course secretion vs intracellular pools, or an inert hydrophobic scaffold control with matched Am/contact angle).
- 9. The comparison to PU foam rafts and aerogels is useful.
1) Emphasize where your triple-mode hybrid ("immobilization" + "elicitation" + "in-situ extraction") adds value beyond PU foam systems already achieving similar YP/X, and what trade-offs (growth penalty, handling, extraction workflow) remain.
Thank you.
Author Response

(The authors gave the same response as above.)

Round 2
Reviewer 2 Report
Comments and Suggestions for Authors
The authors' comprehensive and helpful edits to the manuscript "Hybrid poly(lactic)-chitosan scaffold intensifying in situ bioprocessing of Rindera graeca transgenic roots for enhanced rinderol production" are greatly appreciated. The updated version has been greatly enhanced, and I applaud the authors for their thorough answers and thoughtful evaluation of the reviewer's recommendations. The manuscript's presentation and content have both been significantly improved. I think the manuscript is now ready for publishing following a few minor editorial checks.
Reviewer 3 Report
Comments and Suggestions for Authors
Despite the corrections, the concept of viscosity remains a key one in the manuscript. The authors stated that they would not measure viscosity and removed all references to it from the manuscript. This is not true; the concept of viscosity appears eight times in the manuscript... I find it rather strange that the authors are attempting to so clumsily mislead both the reviewer and potential readers.
The authors believe that the primary goal was not to determine absolute physical density or a standardized value for absorbency, but to compare the relative differences in sorption capacity between the various matrix modifications used in our study (PLA and PLA-chitosan variants). Such statements seem rather odd to me. Science uses the same units of measurement so that experiments can be repeated in different places around the world. Using relative units is harmful and undermines human progress.
I previously wrote that the authors measured the absorption and retention capacity of liquids using oil of unknown viscosity. He also wrote that the ability
of liquids to penetrate cavities depends heavily on the viscosity of the oil and can vary by more than one order of magnitude for different oils. The authors cited oil density instead of viscosity in the manuscript. I would like to point out that these are different physical quantities. Can the authors explain their rationale?
I'm glad the authors corrected the obvious inconsistency with "DB." However, the overall problem remains: the data are presented in relative units, not real ones...
The units h-1 obviously characterize velocity. Can the authors explain the physical meaning or advantage of calculating velocity from a single experimental point?
The manuscript's novelty and technological advancement are insufficient. After reading the work, we learn little new about the fundamental aspects of the process. The main method used in the manuscript is weighing... I also believe that the technological complexity of the work and its level do not correspond to the Q1 journal.